# Which Explanation Should I Choose?
# A Function Approximation Perspective to
# Characterizing Post Hoc Explanations

**Tessa Han**
Harvard University
Cambridge, MA
than@g.harvard.edu

**Suraj Srinivas**
Harvard University
Cambridge, MA
ssrinivas@seas.harvard.edu

**Himabindu Lakkaraju**
Harvard University
Cambridge, MA
hlakkaraju@hbs.edu

## Abstract

A critical problem in the field of post hoc explainability is the lack of a common foundational goal among methods. For example, some methods are motivated by function approximation, some by game theoretic notions, and some by obtaining clean visualizations. This fragmentation of goals causes not only an inconsistent conceptual understanding of explanations but also the practical challenge of not knowing which method to use when.

In this work, we begin to address these challenges by unifying eight popular post hoc explanation methods (LIME, C-LIME, KernelSHAP, Occlusion, Vanilla Gradients, Gradients × Input, SmoothGrad, and Integrated Gradients). We show that these methods all perform local function approximation of the black-box model, differing only in the neighbourhood and loss function used to perform the approximation. This unification enables us to (1) state a *no free lunch theorem for explanation methods*, demonstrating that no method can perform optimally across all neighbourhoods, and (2) provide a *guiding principle* to choose among methods based on faithfulness to the black-box model. We empirically validate these theoretical results using various real-world datasets, model classes, and prediction tasks.

By bringing diverse explanation methods into a common framework, this work (1) advances the conceptual understanding of these methods, revealing their shared local function approximation objective, properties, and relation to one another, and (2) guides the use of these methods in practice, providing a principled approach to choose among methods and paving the way for the creation of new ones.

## 1 Introduction

As machine learning models become increasingly complex and are increasingly deployed in high-stakes settings (e.g., medicine [1], law [2], and finance [3]), there is a growing emphasis on understanding how models make predictions so that decision-makers (e.g., doctors, judges, and loan officers) can assess the extent to which they can trust model predictions. To this end, several post hoc explanation methods have been developed, including LIME [4], C-LIME [5], SHAP [6], Occlusion [7], Vanilla Gradients [8], Gradient x Input [9], SmoothGrad [10], and Integrated Gradients [11]. However, different methods have different goals. Such differences lead to both conceptual and practical challenges to understanding and using explanation methods, thwarting progress in the field.

From a conceptual standpoint, the misalignment of goals among methods leads to an inconsistent view of explanations. What is an explanation? This is unclear as different methods have different notions of explanation. Depending on the method, explanations may be local function approximations (LIME and C-LIME), Shapley values (SHAP), raw gradients (Vanilla Gradients), raw gradients

36th Conference on Neural Information Processing Systems (NeurIPS 2022).

scaled by the input (`Gradient x Input`), de-noised gradients (`SmoothGrad`), or a straight-line path integral of gradients (`Integrated Gradients`). Furthermore, the lack of a common mathematical framework for studying these diverse methods prevents a systematic understanding of these methods and their properties. To address these challenges, this paper unifies diverse explanation methods under a common framework, showing that diverse methods share a common motivation of local function approximation, and uses the framework to investigate and evaluate properties of these methods.

From a practical standpoint, the misalignment of goals among methods leads to the disagreement problem [12], the phenomenon that different methods provide disagreeing explanations for the same model prediction. Not only do different methods often generate disagreeing explanations in practice, but practitioners do not have a principled approach to select among explanations, resorting to ad hoc heuristics such as personal preference [12]. These findings prompt one to ask why explanation methods disagree and how to select among them in a principled manner. This paper addresses these questions, providing both an explanation for the disagreement problem and a principled approach to select among methods.

Thus, to address these conceptual and practical challenges, we study post hoc explanation methods from a function approximation perspective. We formalize a mathematical framework that unifies and characterizes diverse methods and that provides a principled approach to select among methods. Our work makes the following contributions:

1. We show that eight diverse, popular explanation methods (`LIME`, `C-LIME`, `KernelSHAP`, `Occlusion`, `Vanilla Gradients`, `Gradient x Input`, `SmoothGrad`, and `Integrated Gradients`) all perform local function approximation of the black-box model, differing only in the neighbourhoods and loss functions used to perform the approximation.

2. We introduce a *no free lunch theorem for explanation methods* which demonstrates that no single explanation method can perform local function approximation faithfully across all neighbourhoods, which in turn calls for a principled approach to select among methods.

3. To select among methods, we set forth a *guiding principle* based on function approximation, deeming a method to be effective if its explanation recovers the black-box model when the two are in the same model class (i.e., if the explanation perfectly approximates the black-box model when possible).

4. We empirically validate the theoretical results above using various real-world datasets, model classes, and prediction tasks.

## 2   Related Work

**Post hoc explanation methods.** Post hoc explanation methods can be classified based on model access (black-box model vs. access to model internals), explanation scope (global vs. local), search technique (perturbation-based vs. gradient-based), and basic unit of explanation (feature importance vs. rule-based). This paper focuses on local post hoc explanation methods based on feature importance. It analyzes four perturbation-based methods (`LIME`, `C-LIME`, `KernelSHAP`, and `Occlusion`) and four gradient-based methods (`Vanilla Gradients`, `Gradient x Input`, `SmoothGrad`, and `Integrated Gradients`).

**Connections among post hoc explanation methods.** Prior works have taken initial steps towards characterizing post hoc explanation methods and the connections among them. Agarwal et al. [5] proved that `C-LIME` and `SmoothGrad` converge to the same explanation in expectation. Lundberg and Lee [6] proposed a framework based on Shapley values to unify binary perturbation-based explanations. Covert et al. [13] found that many perturbation-based methods share the property of estimating feature importance based on the change in model behavior upon feature removal. In addition, Ancona et al. [14] analyzed four gradient-based explanation methods and the conditions under which they produce similar explanations. However, these analyses are based on mechanistic properties of methods (e.g., Shapley values or feature removal), are limited in scope (connecting only two methods, only perturbation-based methods, or only gradient-based methods), and do not inform when one method is preferable to another. In contrast, this paper formalizes a mathematical framework based on the concept of local function approximation, unifies eight diverse methods (spanning perturbation-based and gradient-based methods), and guides the use of these methods in practice.

**Properties of post hoc explanation methods.** Prior works have examined various properties of post hoc explanation methods, including faithfulness to the black-box model [15–17], robustness to adversarial attack [18–20, 15, 21], and fairness across subgroups [22]. This paper focuses on explanation faithfulness. Related works [15–17] assessed explanations generated by gradient-based methods, finding that they are not always faithful to the underlying model. Different from these works, this paper provides a framework for generating faithful explanations in the first place, theoretically characterizes the faithfulness of existing methods in different input domains, and provides a principled approach to select among methods and develop new ones based on explanation faithfulness.

# 3   Explanation as Local Function Approximation

In this section, we formalize the local function approximation framework and show its connection to existing explanation methods. We start by defining the notation used in the paper.

**Notation.** Let $f : \mathcal{X} \to \mathcal{Y}$ be the black-box function we seek to explain in a post hoc manner, with input domain $\mathcal{X}$ (e.g., $\mathcal{X} = \mathbb{R}^d$ or $\{0,1\}^d$) and output domain $\mathcal{Y}$ (e.g., $\mathcal{Y} = \mathbb{R}$ or $[0,1]$). Let $\mathcal{G} = \{g : \mathcal{X} \to \mathcal{Y}\}$ be the class of interpretable models used to generate a local explanation for $f$ by selecting a suitable interpretable model $g \in \mathcal{G}$.

We characterize locality around a point $\mathbf{x}_0 \in \mathcal{X}$ using a noise random variable $\xi$ which is sampled from distribution $\mathcal{Z}$. Let $\mathbf{x}_\xi = \mathbf{x}_0 \oplus \xi$ be a perturbation of $\mathbf{x}_0$ generated by combining $\mathbf{x}_0$ and $\xi$ using a binary operator $\oplus$ (e.g., addition, multiplication). Lastly, let $\ell(f, g, \mathbf{x}_0, \xi) \in \mathbb{R}^+$ be the loss function (e.g., squared error, cross-entropy) measuring the distance between $f$ and $g$ over the noise random variable $\xi$ around $\mathbf{x}_0$.

We now define the local function approximation framework.

**Definition 1.** *Local function approximation (LFA) of a black-box model $f$ on a neighbourhood distribution $\mathcal{Z}$ around $\mathbf{x}_0$ by an interpretable model class $\mathcal{G}$ and a loss function $\ell$ is given by*

$$g^* = \arg\min_{g \in \mathcal{G}} \; \mathbb{E}_{\xi \sim \mathcal{Z}} \; \ell(f, g, \mathbf{x}_0, \xi) \tag{1}$$

*where a valid loss $\ell$ is such that* $\mathbb{E}_{\xi \sim \mathcal{Z}} \, \ell(f, g, \mathbf{x}_0, \xi) = 0 \iff f(\mathbf{x}_\xi) = g(\mathbf{x}_\xi) \quad \forall \xi \sim \mathcal{Z}$

The LFA framework is a formalization of the function approximation perspective first introduced by LIME [4] to motivate local explanations. Note that this conceptual framework is distinct from the algorithm introduced by LIME. We elaborate on this distinction below.

(1) The LFA framework requires that $f$ and $g$ share the same input domain $\mathcal{X}$ and output domain $\mathcal{Y}$, a fundamental prerequisite for function approximation. This implies, for example, that using an interpretable model g with binary inputs ($\mathcal{X} = \{0,1\}^d$) to approximate a black-box model $f$ with continuous inputs ($\mathcal{X} = \mathbb{R}^d$), as proposed in LIME, is not true function approximation.

(2) By imposing a condition on the loss function, the LFA framework ensures model recovery under specific conditions: $g^*$ recovers $f$ (i.e., $g^* = f$) through LFA when $f$ itself is of the interpretable model class $\mathcal{G}$ (i.e., $f \in \mathcal{G}$) and perturbations span the input domain of $f$ (i.e., domain($\mathbf{x}$) = $\mathcal{X}$). This is a key distinction between the LFA framework and LIME (which has no such requirement) and guides the characterization of explanation methods in Section §4.

(3) Efficiently minimizing Equation 1 requires following standard machine learning methodology of splitting the perturbation data into train / validation / test sets and tuning hyper-parameters on the validation set to ensure generalization. To our knowledge, implementations of LIME do not adopt this procedure, making it possible to overfit to a small number of perturbations.

The LFA framework is generic enough to accommodate a variety of explanation methods. In fact, we show that specific instances of this framework converge to existing methods, as summarized in Table 1. At a high level, existing methods use a linear model $g$ to locally approximate the black-box model $f$ in different input domains (binary or continuous) over different local neighbourhoods specified by noise random variable $\xi$ (where $\xi$ is binary or continuous, drawn from a specified distribution, and combined additively or multiplicatively with point $\mathbf{x}_0$) using different loss functions (squared-error or gradient-matching loss). We discuss the details of these connections in the following sections.

| Explanation Method | Local Neighbourhood $\mathcal{Z}$ around $\mathbf{x}_0$ | Loss Function $\ell$ |
|---|---|---|
| C-LIME | $\mathbf{x}_0 + \xi;\ \xi(\in \mathbb{R}^d) \sim \text{Normal}(0, \sigma^2)$ | Squared Error |
| SmoothGrad | $\mathbf{x}_0 + \xi;\ \xi(\in \mathbb{R}^d) \sim \text{Normal}(0, \sigma^2)$ | Gradient Matching |
| Vanilla Gradients | $\mathbf{x}_0 + \xi;\ \xi(\in \mathbb{R}^d) \sim \text{Normal}(0, \sigma^2), \sigma \to 0$ | Gradient Matching |
| Integrated Gradients | $\xi\mathbf{x}_0;\ \xi(\in \mathbb{R}) \sim \text{Uniform}(0, 1)$ | Gradient Matching |
| Gradients $\times$ Input | $\xi\mathbf{x}_0;\ \xi(\in \mathbb{R}) \sim \text{Uniform}(a, 1), a \to 1$ | Gradient Matching |
| LIME | $\mathbf{x}_0 \odot \xi;\ \xi(\in \{0,1\}^d) \sim \text{Exponential kernel}$ | Squared Error |
| KernelSHAP | $\mathbf{x}_0 \odot \xi;\ \xi(\in \{0,1\}^d) \sim \text{Shapley kernel}$ | Squared Error |
| Occlusion | $\mathbf{x}_0 \odot \xi;\ \xi(\in \{0,1\}^d) \sim \text{Random one-hot vectors}$ | Squared Error |

Table 1: Correspondence of existing explanation methods to instances of the LFA framework. Existing methods perform LFA of a black-box model $f$ using the interpretable model class $\mathcal{G}$ of linear models where $g(\mathbf{x}) = \mathbf{w}^\top \mathbf{x}$ over a local neighbourhood $\mathcal{Z}$ around point $\mathbf{x}_0$ based on a loss function $\ell$. Exponential and Shapley kernels are defined in Appendix A.1.

### 3.1 LFA with Continuous Noise: Gradient-Based Explanation Methods

To connect gradient-based explanation methods to the LFA framework, we leverage the gradient-matching loss function $\ell_{gm}$. We define $\ell_{gm}$ and show that it is a valid loss function for LFA.

$$\ell_{gm}(f, g, \mathbf{x}_0, \xi) = \|\nabla_\xi f(\mathbf{x}_0 \oplus \xi) - \nabla_\xi g(\mathbf{x}_0 \oplus \xi)\|_2^2 \qquad (2)$$

This loss function has been previously used in the contexts of generative modeling (where it is dubbed score-matching) [23] and model distillation [16]. However, to our knowledge, its use in interpretability is novel.

**Proposition 1.** *The gradient-matching loss function $\ell_{gm}$ is a valid loss function for LFA up to a constant, i.e., $\mathbb{E}_{\xi \sim \mathcal{Z}}\, \ell_{gm}(f, g, \mathbf{x}_0, \xi) = 0 \iff f(\mathbf{x}_\xi) = g(\mathbf{x}_\xi) + C \quad \forall \xi \sim \mathcal{Z}$, where $C \in \mathbb{R}$.*

*Proof.* If $f(\mathbf{x}_\xi) = g(\mathbf{x}_\xi)$, then $\nabla_\xi f(\mathbf{x}_\xi) = \nabla_\xi g(\mathbf{x}_\xi)$ and it follows from the definition of $\ell_{gm}$ that $\ell_{gm} = 0$. Integrating $\nabla_\xi f(\mathbf{x}_\xi) = \nabla_\xi g(\mathbf{x}_\xi)$ gives $f(\mathbf{x}_\xi) = g(\mathbf{x}_\xi) + C$. $\qquad\square$

Proposition 1 implies that, when using the linear model class $\mathcal{G}$ parameterized by $g(\mathbf{x}) = \mathbf{w}^\top \mathbf{x} + b$ to approximate $f$, $g^*$ recovers $\mathbf{w}$ but not $b$. This can be fixed by setting $b = f(0)$.

**Theorem 1.** *LFA with gradient-matching loss is equivalent to (1) `SmoothGrad` for additive continuous Gaussian noise, which converges to `Vanilla Gradients` in the limit of a small standard deviation for the Gaussian distribution; and (2) `Integrated Gradients` for multiplicative continuous Uniform noise, which converges to `Gradient x Input` in the limit of a small support for the Uniform distribution.*

*Proof Sketch.* For `SmoothGrad` and `Integrated Gradients`, the idea is that these methods are exactly the first-order stationary points of the gradient-matching loss function under their respective noise distributions. In other words, the weights of the interpretable model $g$ that minimize the loss function is the explanation returned by each method. For `Vanilla Gradients` and `Gradient x Input`, the result is derived by taking the specified limits and using the Dirac delta function to calculate the limit. In the limit, the weights of the interpretable model $g$ converge to the explanation of each method. The full proof is in Appendix A.1.

Along with gradient-based methods, `C-LIME` (a perturbation-based method) is an instance of the LFA framework by definition, using the squared-error loss function. The analysis in this section characterizes methods that use continuous noise. It does not extend to binary or discrete noise methods because gradients and continuous random variables do not apply in these domains. In the next section, we discuss binary noise methods.

### 3.2 LFA with Binary Noise: LIME, KernelSHAP and Occlusion maps

**Theorem 2.** *LFA with multiplicative binary noise and squared-error loss is equivalent to (1) LIME for noise sampled from an unnormalized exponential kernel over binary vectors; (2) KernelSHAP*

*for noise sampled from an unnormalized Shapley kernel; and (3) Occlusion for noise in the form of one-hot vectors.*

*Proof Sketch.* For `LIME` and `KernelSHAP`, the equivalence is mostly by definition: these methods have components that correspond to the interpretable model $g$ and the loss function $\ell$ of the LFA framework and we need only to determine the local neighbourhood $\mathcal{Z}$. We define the local neighbourhood $\mathcal{Z}$ using each method's weighting kernel. In this setup, the LFA framework yields the respective explanation methods in expectation via importance sampling. For `Occlusion`, the equivalence involves enumerating all perturbations, specifying an appropriate loss function, and computing the resulting stationary points of the loss function. The full proof is in Appendix A.1.

### 3.3 Which Methods Do Not Perform LFA?

Some popular explanation methods are not instances of the LFA framework due to their properties. These methods include guided backpropagation [24], DeconvNet [25], Grad-CAM [26], Grad-CAM++ [27], FullGrad [28], and DeepLIFT [9]. Further details are in Appendix A.2.

## 4 When Do Explanations Perform Model Recovery?

Having described the LFA framework and its connections to existing explanation methods, we now leverage this framework to analyze the performance of methods under different conditions. We introduce a *no free lunch theorem for explanation methods*, inspired by classical no free lunch theorems in learning theory and optimization. Then, we assess the ability of existing methods to perform *model recovery* based on which we provide recommendations for choosing among methods.

### 4.1 No Free Lunch Theorem for Explanation Methods

An important implication of the function approximation perspective is that no explanation can be optimal across all neighbourhoods because each explanation is designed to perform LFA in a specific neighbourhood. This is especially true for explanations of non-linear models. We formalize this intuition into the following theorem.

**Theorem 3** (No Free Lunch for Explanation Methods). *Consider explaining a black-box model $f$ around point $\mathbf{x}_0$ using an interpretable model $g$ from model class $\mathcal{G}$ and a valid loss function $\ell$ where the distance between $f$ and $\mathcal{G}$ is given by $d(f, \mathcal{G}) = \min_{g \in \mathcal{G}} \max_{\mathbf{x} \in \mathcal{X}} \ell(f, g, 0, \mathbf{x})$.*

*Then, for any explanation $g^*$ over a neighbourhood distribution $\xi_1 \sim \mathcal{Z}_1$ such that $\max_{\xi_1} \ell(f, g^*, \mathbf{x}_0, \xi_1) \leq \epsilon$, there always exists another neighbourhood $\xi_2 \sim \mathcal{Z}_2$ such that $\max_{\xi_2} \ell(f, g^*, \mathbf{x}_0, \xi_2) \geq d(f, \mathcal{G})$.*

*Proof Sketch.* The idea is that, given an explanation obtained by using $g$ to approximate $f$ over a specific local neighbourhood $\mathcal{Z}$, it is always possible to find a local neighbourhood over which this explanation does not perform well (i.e., does not perform faithful LFA). Thus, no single explanation method can perform well over all local neighbourhoods. The proof entails constructing an "adversarial" input for an explanation $g^*$ such that $g^*$ has a large loss for this input and then creating a neighbourhood that contains this adversarial input which will provably have a large loss. The magnitude of this loss is $d(f, \mathcal{G})$, the distance between $f$ and the model class $\mathcal{G}$, inspired by the Haussdorf distance. The proof is generic and makes no assumptions regarding the forms of $\ell$, $\mathcal{G}$ or $\mathcal{Z}_1$. The full proof is in Appendix A.3.

Thus, an explanation on a finite $\mathcal{Z}_1$ necessarily cannot approximate function behaviour at all other points, especially when $\mathcal{G}$ is less expressive than $f$, which is indicated by a large value of $d(f, \mathcal{G})$. Thus, in the general case, one cannot perform model recovery as $\mathcal{G}$ is less expressive than $f$.

An important implication of Theorem 3 is that seeking to find the "best" explanation without specifying a corresponding neighbourhood is futile as no universal "best" explanation exists. Furthermore, once the neighbourhood is specified, the best explanation is exactly the one given by the corresponding instance of the LFA framework.

In the next section, we consider the special case when $d(f, \mathcal{G}) = 0$ (i.e., when $f \in \mathcal{G}$), where Theorem 3 does not apply because the same explanation can be optimal for multiple neighbourhoods and model recovery is thus possible.

## 4.2 Characterizing Explanation Methods via Model Recovery

Next, we formally state the model recovery condition for explanation methods. Then, we use this condition as a guiding principle to choose among methods.

**Definition 2** (Model Recovery: Guiding Principle). *Given an instance of the LFA framework with a black-box model $f$ such that $f \in \mathcal{G}$ and a specific noise type (e.g., Gaussian, Uniform), an explanation method performs model recovery if there exists some noise distribution $\mathcal{Z}$ such that LFA returns $g^* = f$.*

In other words, when the black-box model $f$ itself is of the interpretable model class $G$, there must exist some setting of the noise distribution (within the noise type specified in the instance of the LFA framework) that is able to recover the black-box model. Thus, in this special case, we require *local function approximation* to lead to *global model recovery* over all inputs. This criterion can be thought of as a "sanity check" for explanation methods to ensure that they remain faithful to the black-box model.

Next, we analyze the impact of the choice of perturbation neighbourhood $\mathcal{Z}$, the binary operator $\oplus$, and the interpretable model class $\mathcal{G}$ on an explanation method's ability to satisfy the model recovery guiding principle in different input domains $\mathcal{X}$. Note that while we can choose $\mathcal{Z}$, $\oplus$, and $\mathcal{G}$, we cannot choose $\mathcal{X}$, the input domain.

**Which explanation should I choose for continuous $\mathcal{X}$?** We now analyze the model recovery properties of existing explanation methods when the input domain is continuous. We consider methods based on additive continuous noise (`SmoothGrad`, `Vanilla Gradients`, and `C-LIME`), multiplicative continuous noise (`Integrated Gradients` and `Gradient x Input`), and multiplicative binary noise (`LIME`, `KernelSHAP`, and `Occlusion`). For these methods, we make the following remark regarding model recovery for the class of linear models.

**Remark 1.** *For $\mathcal{X} = \mathbb{R}^d$ and linear models $f$ and $g$ where $f(\mathbf{x}) = \mathbf{w}_f^\top \mathbf{x}$ and $g(\mathbf{x}) = \mathbf{w}_g^\top \mathbf{x}$, additive continuous noise methods recover $f$ (i.e., $\mathbf{w}_g = \mathbf{w}_f$) while multiplicative continuous and multiplicative binary noise methods do not and instead recover $\mathbf{w}_g = \mathbf{w}_f \odot \mathbf{x}$.*

This remark can be verified by directly evaluating the explanations (weights) of linear models, where the gradient exactly corresponds to the weights.

Note that the inability of multiplicative continuous noise methods to recover the black-box model is not due to the multiplicative nature of the noise, but due to the parameterization of the loss function. Specifically, these methods (implicitly) use the loss function $\ell(f, g, \mathbf{x}_0, \xi) = \|\nabla_\xi f(\mathbf{x}_\xi) - \nabla_\xi g(\xi)\|_2^2$. Slightly changing the loss function to $\ell(f, g, \mathbf{x}_0, \xi) = \|\nabla_\xi f(\mathbf{x}_\xi) - \nabla_\xi g(\mathbf{x}_\xi)\|_2^2$, i.e., replacing $g(\xi)$ with $g(\mathbf{x}_\xi)$, would enable $g^*$ to recover $f$. This would change `Integrated Gradients` to $\int_{\alpha=0}^1 \nabla_{\alpha\mathbf{x}} f(\alpha\mathbf{x})$ (omitting the input multiplication term) and `Gradient x Input` to `Vanilla Gradients`.

A similar argument can be made for binary noise methods which parameterize the loss function as $\ell(f, g, \mathbf{x}_0, \xi) = \|f(\mathbf{x}_\xi) - g(\xi)\|^2$. By changing the loss function to $\ell(f, g, \mathbf{x}_0, \xi) = \|f(\mathbf{x}_\xi) - g(\mathbf{x}_\xi)\|^2$, binary noise methods can recover $f$ for the case described in Remark 1. However, binary noise methods for continuous domains are unreliable, as there are cases where, despite the modification to $\ell$, model recovery is not guaranteed. The following is an example of this scenario.

**Remark 2.** *For $\mathcal{X} = \mathbb{R}^d$, periodic functions $f$ and $g$ where $f(\mathbf{x}) = \sum_{i=1}^d \sin(\mathbf{w}_{f_i} \odot \mathbf{x}_i)$ and $g(\mathbf{x}) = \sum_{i=1}^d \sin(\mathbf{w}_{g_i} \odot \mathbf{x}_i)$, and an integer $n$, binary noise methods do not perform model recovery for $|w_{f_i}| \geq \frac{n\pi}{\mathbf{x}_{0_i}}$.*

This is because, for the conditions specified, $\sin(\mathbf{w}_{f_i} \mathbf{x}_{0_i}) = \sin(\pm n\pi) = \sin(0) = 0$, i.e., $\sin(\mathbf{w}_{f_i} \mathbf{x}_{0_i})$ outputs zero for all binary perturbations, thereby preventing model recovery. In this case, the discrete nature of the noise makes model recovery impossible. In general, discrete noise is inadequate for the recovery of models with large frequency components.

**Which explanation should I choose for binary $\mathcal{X}$?** In the binary domain, continuous noise methods are invalid, restricting the choice of methods to binary noise methods. For reasons discussed above, methods with perturbation neighbourhoods characterized by multiplicative binary perturbations (e.g., `LIME`, `KernelSHAP`, and `Occlusion`) only enable $g^*$ to recover $f$ in the binary domain. Note that

the sinusoidal example in Remark 2 does not apply in this regime due to the continuous nature of its domain.

**Which explanation should I choose for discrete $\mathcal{X}$?** In the discrete domain, continuous noise methods are also invalid. In addition, binary noise methods (e.g., LIME, KernelSHAP and Occlusion) cannot be used either because model recovery is not guaranteed in the sinusoidal case (Remark 2), following similar logic to that presented for continuous noise. Note that none of the existing methods in Table 1 perform general discrete perturbations, suggesting that these methods are not suitable for the discrete domain. Thus, in the discrete domain, a user can apply the LFA framework to define a new explanation method, specifying an appropriate discrete noise type. In the next section, we discuss more broadly about how one can use the LFA framework to create novel explanation methods.

### 4.3 Designing Novel Explanations with LFA

The LFA framework not only unifies existing explanation methods but also guides the creation of new ones. To explain a given black-box model prediction using the LFA framework, a user must specify the (1) interpretable model class $\mathcal{G}$, (2) neighbourhood distribution $\mathcal{Z}$, (3) loss function $\ell$, and (4) binary operator $\oplus$ to combine the input and the noise. Specifying these four components completely specifies an instance of the LFA framework, thereby generating an explanation method tailored to a given context.

To illustrate this, consider a scenario in which a user seeks to create a sparse variant of SmoothGrad that yields non-zero gradients for only a small number of features ("SparseSmoothGrad"). Designing SparseSmoothGrad only requires the addition of a regularization term to the loss function used in the SmoothGrad instance of the LFA framework (e.g., $\ell = \ell_{SmoothGrad} + \|\nabla_\xi g(\mathbf{x}_\xi)\|_0$), at which point, sparse solvers may be employed to solve the problem. Note that, unlike SmoothGrad, SparseSmoothGrad does not have a closed form solution, but that is not an issue for the LFA framework. More generally, by allowing customization of (1), (2), (3), and (4), the LFA framework creates new explanation methods through "variations on a theme".

We summarize Section §4 as a table in Appendix A.4 and discuss the practical implications of Section §4 by providing the following recommendation for choosing among explanation methods.

**Recommendation for choosing among explanation methods.** In general, choose methods that satisfy the guiding principle of model recovery in the input domain in question. For continuous data, use additive continuous noise methods (e.g., SmoothGrad, Vanilla Gradients, C-LIME) or modified multiplicative continuous noise methods (e.g., Integrated Gradients, Gradient x Input) as described in Section §4.2. For binary data, use binary noise methods (e.g., LIME, KernelSHAP, Occlusion). Given that methods that use discrete noise do not exist, in case of discrete data, design novel explanation methods using the LFA framework with discrete noise neighbourhoods. Within each input domain, choosing among appropriate methods boils down to determining the perturbation neighbourhood most suitable in the given context.

## 5 Empirical Evaluation

In this section, we present an empirical evaluation of the LFA framework. We first describe the experimental setup and then discuss three experiments and their findings.

### 5.1 Datasets, Models, and Metrics

**Datasets.** We experiment with two real-world datasets for two prediction tasks. The first dataset is the life expectancy dataset from the World Health Organization (WHO) [29]. It consists of countries' demographic, economic, and health factors from 2000 to 2015, with 2,938 observations for 20 continuous features. We use this dataset to perform regression, predicting life expectancy. The other dataset is the home equity line of credit (HELOC) dataset from FICO [30]. It consists of information on HELOC applications, with 9,871 observations for 24 continuous features. We use this dataset to perform classification, predicting whether an applicant made payments without being 90 days overdue. Additional dataset details are described in Appendix A.5.

**Models.** For each dataset, we train four models: a simple model (linear regression for the WHO dataset and logistic regression for the HELOC dataset) that can satisfy conditions of the guiding prin-

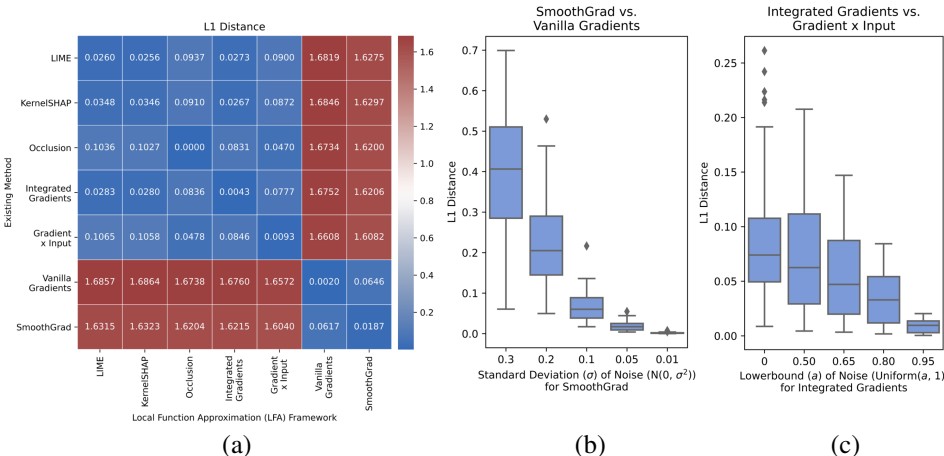

(a)               (b)               (c)

Figure 1: Correspondence between existing explanation methods and instances of the LFA framework. (a) Heatmap of average L1 distance between pairs of explanations. Boxplots of L1 distance between explanations of (b) `SmoothGrad` and `Vanilla Gradients` and (c) `Integrated Gradients` and `Gradient x Input`. The lower the L1 distance, the more similar two explanations are. Results indicate that existing explanation methods are instances of the LFA framework.

ciple and three more complex models (neural networks of varying complexity) that are more reflective of real-world applications. Model architectures and performance are described in Appendix A.5.

**Metrics.** To measure the similarity between two vectors (e.g., between two sets of explanations or between an explanation and the true model weights), we use L1 distance and cosine distance. L1 distance ranges between $[0, \infty)$ and is 0 when two vectors are the same. Cosine distance ranges between $[0, 2]$ and is 0 when the angle between two vectors is $0°$ (or $360°$). For both metrics, the lower the value, the more similar two given vectors are.

## 5.2 Experiments

Here, we describe the setup of the experiments, present results, and discuss their implications.

**Experiment 1: Existing explanation methods are instances of the LFA framework.** First, we compare existing methods with corresponding instances of the LFA framework to assess whether they generate the same explanations. To this end, we use seven methods to explain the predictions of black-box models for 100 randomly-selected test set points. For each method, explanations are computed using either the existing method (implemented by Meta's Captum library [31]) or the corresponding instance of the LFA framework (Table 1). The similarity of a given pair of explanations is measured using L1 distance and cosine distance.

The L1 distance values for a neural network with three hidden layers trained on the WHO dataset are shown in Figure 1. In Figure 1a, lowest L1 distance values appear in the diagonal of the heatmap, indicating that explanations generated by existing methods and corresponding instances of the LFA framework are very similar. Figures 1b and 1c show that explanations generated by instances of the LFA framework corresponding to `SmoothGrad` and `Integrated Gradients` converge to those of `Vanilla Gradients` and `Gradient x Input`, respectively. Together, these results demonstrate that, consistent with the theoretical results derived in Section §3, existing methods are instances of the LFA framework. In addition, the clustering of the methods in Figure 1a indicates that, consistent with the theoretical analysis in Section §4, for continuous data, `SmoothGrad` and `Vanilla Gradients` generate similar explanations while `LIME`, `KernelSHAP`, `Occlusion`, `Integrated Gradients`, and `Gradient x Input` generate similar explanations. We observe similar results across various datasets, models, and metrics (Appendix A.6.1).

**Experiment 2: Some methods recover the underlying model while others do not (guiding principle).** Next, we empirically assess which existing methods satisfy the guiding principle, i.e., which methods recover the black-box model $f$ when $f$ is of the interpretable model class $\mathcal{G}$. We specify a setting in which $f$ and $g$ are of the same model class, generate explanations using each method, and assess whether $g$ recovers $f$ for each explanation. For the WHO dataset, we set $f$

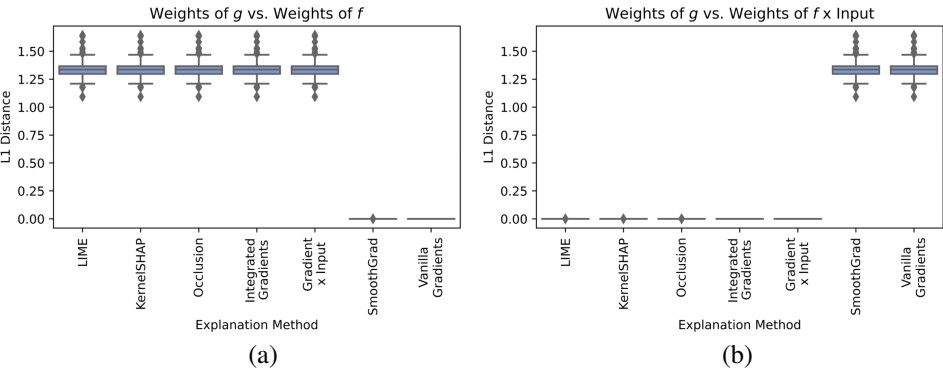

(a)             (b)

Figure 2: Analysis of model recovery. The lower the L1 distance, the more similar $g$'s weights are to (a) $f$'s weights or (b) $f$'s weights multiplied by the input. Results indicate that, for continuous data, additive continuous noise methods recover $f$'s weights, satisfying the guiding principle, while multiplicative binary and multiplicative continuous noise methods do not, recovering $f$'s weights multiplied by the input instead.

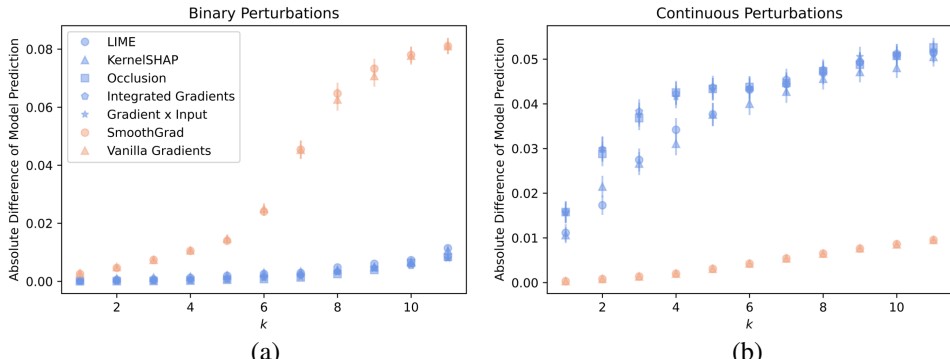

(a)             (b)

Figure 3: Perturbation tests perturbing bottom $k$ features using (a) binary or (b) continuous noise. The lower the curve, the better a method identifies unimportant features. Results illustrate the no free lunch theorem, i.e., no single method performs best across all neighborhoods.

and $g$ to be linear regression models and generate explanations for 100 randomly-selected test set points. Then, for each point, we compare $g$'s weights with $f$'s gradients alone or with $f$'s gradients multiplied by the input because, based on Section §4, some methods generate explanations on the scale of gradients while others on the scale of gradient-times-input. Note that, for linear regression, $f$'s gradients are $f$'s weights.

Results are shown in Figure 2. Consistent with Section §4, for continuous data, `SmoothGrad` and `Vanilla Gradients` recover the black-box model, thereby satisfying the guiding principle, while `LIME`, `KernelSHAP`, `Occlusion`, `Integrated Gradients`, and `Gradient x Input` do not. We observe similar results for the HELOC dataset using logistic regression models for $f$ and $g$ (Appendix A.6.2).

**Experiment 3: No single method performs best across all neighbourhoods (no free lunch theorem).** Lastly, we perform a set of experiments to illustrate the no free lunch theorem in Section §4. We generate explanations for black-box model predictions for 100 randomly-selected test set points and evaluate the explanations using perturbation tests based on top-$k$ or bottom-$k$ features. For perturbation tests based on top-$k$ features, the setup is as follows. For a given data point, $k$, and explanation, we identify the top-$k$ features and either replace them with zero (binary perturbation) or add Gaussian noise to them (continuous perturbation). Then, we calculate the absolute difference in model prediction before and after perturbation. For each point, we generate one binary perturbation (since such perturbations are deterministic) and 100 continuous perturbations (since such perturbations are random), computing the average absolute difference in model prediction for the latter. In this setup, methods that better identify important features yield larger changes in model prediction. For perturbation tests based on bottom-$k$ features, we follow the same procedure

but perturb the bottom-$k$ features instead. In this setup, methods that better identify unimportant features yield smaller changes in model prediction.

Results of perturbation tests based on bottom-$k$ features performed on explanations for a neural network with three hidden layers trained on the WHO dataset are displayed in Figure 3. Consistent with the no free lunch theorem in Section §4, LIME, KernelSHAP, Occlusion, Integrated Gradients, and Gradient x Input perform best on binary perturbation neighbourhoods (Figure 3a) while SmoothGrad and Vanilla Gradients perform best on continuous perturbation neighborhoods (Figure 3b). We observe consistent results across perturbation test types (top-$k$ and bottom-$k$), datasets, and models (Appendix A.6.3). These results have important implications: one should carefully consider the perturbation neighborhood not only when selecting a method to generate explanations but also when selecting a method to evaluate explanations. In fact, the type of perturbations used to evaluate explanations directly determines explanation method performance.

## 6  Conclusions and Future Work

In this work, we formalize the *local function approximation (LFA)* framework and demonstrate that eight popular explanation methods can be characterized as instances of this framework with different local neighbourhoods and loss functions. We also introduce the *no free lunch theorem for explanation methods*, showing that no single method can perform optimally across all neighbourhoods, and provide a *guiding principle* for choosing among methods.

The function approximation perspective captures the essence of an explanation – a simplification of the real world (i.e., a black-box model) that is nonetheless accurate enough to be useful (i.e., predict outcomes of a set of perturbations). When the real world is "simple", an explanation should completely capture its behaviour, a hallmark expressed precisely by the guiding principle. When the requirements of two explanations are distinct (i.e., they are trained to predict different sets of perturbations), then the explanations are each accurate in their own domain and may disagree, a phenomenon captured by the no free lunch theorem.

Our work makes fundamental contributions. We *unify* popular explanation methods, bringing diverse methods into a common framework. Unification brings *conceptual coherence and clarity*: diverse explanation methods, even those seemingly unrelated to function approximation, perform LFA but differ in the way they perform it. Unification also enables *theoretical simplicity*: to study diverse explanation methods, instead of analyzing each method individually, one can simply analyze the LFA framework and apply the findings to each method. An example of this is the no free lunch theorem which holds true for all instances of the LFA framework. Furthermore, our work provides *practical guidance* by presenting a principled approach to select among methods and design new ones.

Our work also addresses key open questions in the field. In response to criticism about the lack of consensus in the field regarding the overarching goals of post hoc explainability [32], our work points to function approximation as a principled goal. It also provides an explanation for the disagreement problem [12], i.e., why different methods generate different explanations for the same model prediction. According to the LFA framework, this disagreement occurs because different methods approximate the black-box model over different neighbourhoods using different loss functions.

Future research includes the following directions. First, we analyzed eight popular post hoc explanation methods and this analysis could be extended to other methods. Second, our work focuses on the faithfulness rather than interpretability of explanations. The latter is encapsulated in the "interpretable" model class $\mathcal{G}$, which includes all the information about human preferences with regards to interpretability. However, it is unclear what constitutes an interpretable explanation and elucidating this takes not only conceptual understanding but also human-computer interaction research such as user studies. These are important directions for future research.

## Acknowledgements

The authors would like to thank the anonymous reviewers for their helpful feedback and the following funding agencies for supporting this work. This work is supported in part by NSF awards #IIS-2008461 and #IIS-2040989, and research awards from Google, JP Morgan, Amazon, Harvard Data Science Initiative, and Dˆ3 Institute at Harvard. H.L. would like to thank Sujatha and Mohan Lakkaraju for their continued support and encouragement. T.H. is supported in part by an NSF GRFP fellowship. The views expressed here are those of the authors and do not reflect the official policy or position of the funding agencies.

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
