# OpenReview forum: "Which Explanation Should I Choose? A Function Approximation Perspective to Characterizing Post Hoc Explanations"
_NeurIPS.cc/2022/Conference — NeurIPS 2022 Accept_

### Official Review · Reviewer_JzYe · 2022-07-06

**Rating:** 6
**Confidence:** 2
**Soundness:** 3 good
**Presentation:** 3 good
**Contribution:** 3 good

**Summary:**

The authors propose the local function approximation (LFA) framework and prove that numerous local explanations are an instance of this framework. In their proposed no-free lunch theorem of explanations, the authors show that no single explanation can outperform the rest of the explanations in all neighbors around the local instances. The claims are shown across numerous datasets and models.

Overall, I vote for the acceptance of the paper. I do believe that the proposed approach is sound and interesting. However, I suspect that some of the assumptions in the paper need more careful refinement.

I am open to changing my score in case the authors can provide sound answers to the questions I have written in the questions section.

**Questions:**

* I agree with the authors that the loss formulation of LFA resembles LIME (Equation 1). But we also know that LIME does much. LIME does feature selection before training the surrogate model. Can the authors explain whether the claims and findings still apply?

* Can the authors provide a similar visualization as Figure 3 to show the permutation tests for cases where important features were removed? I think it is important to understand the model recovery in those cases as well.

**Limitations:**

Line 364-370: The authors provide some limitations of their work about including a subset of all explanation techniques. As a reader, I was keen to know a more detailed discussion on 1) limitations of their proposed framework, e.g. where LFA formulation cannot capture explanation techniques 2) Limitations of the assumptions in LFA framework for some application scenarios.

**Strengths And Weaknesses:**

Strength:

* The problem of unifications explanation techniques is an essential and important topic of study
* The authors have built upon numerous other related works and include formalism to strengthen their arguments.
* The evaluation includes numerous explanation techniques and datasets

Limitations:

* The paper is dense and was very hard for me to read. I kindly suggest to the authors to try to help readers navigate all the theories in their work by including more explanations and not relying on compact forms of writing as theorems (especially in Section 3)

---

> ### Author Response · Authors · 2022-08-02
> **Response to Reviewer JzYe - Part 1**
>
> We thank the reviewer for their detailed comments. Below, we address the questions and comments raised by the reviewer.
>
>
> **additional explanations for theorems**:  We really appreciate your effort in reading our paper, and we’re sorry that you found it difficult to navigate. In response to your comment, we will incorporate more intuition and explanation in the final paper, to complement the theorems.
>
> **LIME performs feature selection, is that incorporated in LFA?**: Yes, our claims still apply. While it’s true that in some situations, LIME performs feature selection as a preprocessing step, this is equivalent to adding a regularization term to its loss function (L0 norm). This version of LIME is still an instance of the LFA framework. To formulate this version of LIME as an instance of the LFA framework, we can take the instance of the LFA framework corresponding to LIME (specified in Table 1) and modify the loss function (by adding the L0 norm regularizer), thereby incorporating the feature selection step that LIME does. More broadly, if a method uses regularization, we can still formulate this method as an instance of the LFA framework by adding the same regularization to the loss function component of the LFA framework.
>
>
> **Provide similar visualization as Figure 3 to show cases where important features were removed**: Thank you for the suggestion. We performed the requested set of experiments. The results are attached in Section 2 of “additional_results_rebuttal.pdf” (Supplementary Materials). We will include these experiments in the final paper.
>
> Setup: In Figure 3 in the paper, perturbation tests perturb the bottom-k features and lower curves indicate better performance of a method (rationale: bottom-k features are unimportant → perturbing these features should not change model output by much → lower curve indicates method better identifies unimportant features). In the main paper, following the logic of prior works (lines 331-332), we focused on perturbing the bottom-k features to disentangle the effect of perturbations and true feature importance (measured by top-k features) on the change in model output. This logic is explained in more detail in Section 5.1 of Reference 30. In the new set of experiments performed following the reviewer’s recommendation, perturbation tests perturb the top-k features and higher curves indicate better performance of a method (rationale: top-k features are important → perturbing these features should change model output by a lot → higher curve indicates method better identifies important features).
>
> Results: The original experiments in the paper (Figures A, B, and C) and newly-performed experiments yield consistent results and reinforce our initial conclusion: when perturbations use binary noise, the blue methods (LIME, KernelSHAP, Occlusion, Integrated Gradients, and Gradient x Input) outperform the orange methods (SmoothGrad and Vanilla Gradients), and when perturbations use continuous noise, the orange methods outperform the blue methods.
>
> Thus, the results from existing and new experiments indicate that, consistent with the no free lunch theorem, no single method can perform best across all perturbation neighborhoods. This finding holds across models (linear regression, logistic regression, neural networks) and tasks (regression and classification).
>
>
> **Limitations of LFA: When can LFA not capture explanation techniques?**: Methods such as guided backpropagation, DeconvNet, Grad-CAM, Grad-CAM++, FullGrad, and DeepLIFT cannot be written as LFAs. For details and explanations regarding this, please see Appendix A.1. As the respective sections argue, this is less a limitation of LFA and are more limitations of these methods which are either (1) independent of the black box model they intend to explain, or (2) dependent on the black box model parameterization rather than their functionality, both of which are incompatible with LFA, which requires explanations to (1) depend on the black box model, (2) depend on only the functional form of the black box model.
>
> A genuine limitation of LFA currently is that it can only capture feature attribution methods, and methods such as counterfactual explanations, influence function explanations, anchor explanations, etc that are distinct from feature attribution cannot yet be written as LFAs.
>
>
> (continued)

---

> > ### Author Response · Authors · 2022-08-02
> > **Response to Reviewer JzYe - Part 2**
> >
> > **Limitations of LFA assumptions for applications** : LFA involves first identifying (1) an interpretable model family, (2) a perturbation neighborhood, (3) loss function. For a given application, it may be difficult to identify these correctly. In this paper, we have completely omitted the question of how to choose (1) the interpretable family, while we have partially explored the question of how to choose (2) perturbation neighborhood or (3) loss function in accordance with the guiding principle. An exploration of how to choose these quantities is an interesting topic for future work - as it would automatically answer the question of “which explanation to choose” in a problem dependent manner.
> >
> > We hope that our response addresses your questions/comments and we kindly request you to consider increasing your score. Thank you very much!

---

> > > ### Comment · Reviewer_JzYe · 2022-08-08
> > > **Response of Reviewer JzYe**
> > >
> > > I would like to thank the authors for answering the questions in my review.
> > >
> > > In the proposed approach, your aim to formulate different explanation techniques into LFA and then draw certain conclusions about their ability to perform model recovery. However, the main focus of your work is on the effect of "sampling". However, the explanation techniques you have used follow a pipeline in which sampling is only one stage of that pipeline. I recommend that you aim at investigating other stages of the pipeline such as feature selection and so forth. In addition, the effect of the chosen surrogate model in each explanation is very crucial as well.
> > >
> > > What I meant by the lack of studying the limitation of the proposed approach: for example, you are studying the model recovery on a linear or a sinus curve (Remark 1 and 2). Why focus on these types of models? What happens if we study the model recovery on an XOR or a polynomial function, etc? In addition, what is not studied when we analyze the explanations method in the LFA framework?
> > >
> > > Lastly, thank you for including the study of removing the top-k essential features. I agree that it is consistent with the paper's conclusions.

---

> > > > ### Author Response · Authors · 2022-08-08
> > > > **Follow Up Response to New Comments from Reviewer JzYe (Part 1)**
> > > >
> > > > Thank you for your response, and for engaging with us during the discussion period.
> > > >
> > > > **The main focus of your work is on the effect of "sampling”; what about investigating feature selection and the effect of the chosen surrogate model?**
> > > >
> > > > Note that different ways of **sampling** perturbations around an instance correspond to different definitions of local neighborhoods. Since our goal is to characterize the local neighborhoods and loss functions employed by various state-of-the-art explanation methods, and leverage these characterizations to theoretically and empirically analyze the faithfulness of these methods (i.e., how well each method recovers the underlying model), it might appear that our main focus is on the effect of “sampling”. Furthermore, several *prior works have argued that one of the key drawbacks of local explanation methods is that the notion of local neighborhood (i.e., the exact sampling process) employed by each of these methods is not well understood and/or is ill-defined* [1, 2, 3]. Our research and more specifically our LFA framework addresses this fundamental problem by providing a clear specification of the local neighborhoods employed by various popular state-of-the-art explanation methods.
> > > >
> > > > In addition to analyzing different definitions of local neighborhoods (aka sampling methods), we also consider different loss functions (e.g., mean squared error loss, gradient matching loss etc.) in our work.
> > > >
> > > > Regarding **feature selection** — Note that our framework also accounts for the feature selection aspect, albeit, somewhat implicitly. More specifically, the LFA framework accounts for feature selection by accommodating it in the loss functions (See Table 1 in the main paper). For instance, LIME’s objective function employs a L0 regularization [4], and this is operationalized by first carrying out a feature selection preprocessing step for choosing a subset of important features, and then fitting a linear model only on these features. We can include the L0 regularization term in the loss functions in Table 1 to accommodate the effect of feature selection, and all the resulting functions would still remain instances of the LFA framework, and our theoretical results (e.g., no free lunch theorem for explanation methods) would also hold on these regularized loss functions.
> > > >
> > > > Regarding **surrogate models** – Note that our work is the first to characterize the local neighborhoods and loss functions employed by various state-of-the-art explanation methods, and leverage these characterizations to theoretically and empirically analyze the faithfulness of these methods(i.e., how well each method recovers the underlying model) under various conditions (e.g., varying input domains). In doing so, we already consider multitude of factors -- e.g., different definitions of local neighborhoods (e.g., binary vs. continuous perturbations, additive vs. multiplicative perturbations), different loss functions (e.g., mean squared error loss, gradient matching loss), different input domains (e.g., binary vs. continuous vs. discrete variables). The aforementioned diverse factors already make our theoretical and empirical analysis quite complex and involved. Furthermore, our theoretical analysis (Theorems 1 and 2, Section 3 in the main paper) clearly shows that popular explanation methods (e.g., Gradient based methods, LIME, KernelSHAP) are all performing local linear approximations of the underlying models. Given the above and the fact that this is the first work to theoretically characterize the faithfulness of various post hoc explanation methods, we decided to focus our analysis on linear model explanations/approximations and leave the analysis with other types of surrogate models for future work.
> > > >
> > > > **You are studying the model recovery on a linear or a sinus curve (Remark 1 and 2). Why focus on these types of models?**
> > > >
> > > > In Section 4.2, we were merely trying to find simple examples (or counterexamples) of underlying models which state-of-the-art explanation methods cannot recover. We found that some of the state-of-the-art local explanation methods are unable to recover even simple linear or sinusoidal functions, and included these in our remarks 1 and 2. These functions, in fact, represent the *simplest classes of black box models* which cannot be recovered by several existing explanation methods. We believe that starting our analysis from such simple functions is critical to gain intuition and insight about the behavior of explanation methods.
> > > >
> > > > [Continued in Part 2 below]

---

> > > > > ### Author Response · Authors · 2022-08-08
> > > > > **Follow Up Response to New Comments from Reviewer JzYe (Part 2)**
> > > > >
> > > > > **What happens if we study the model recovery on an XOR or a polynomial function, etc?**
> > > > > Our analysis informs what happens in the case of XOR or polynomials as well. For instance, we expect additive binary perturbation methods (e.g., LIME, KernelSHAP, Occlusion) to perform model recovery for XOR as they are able to generate binary perturbations required to enumerate XOR, and additive continuous noise methods (e.g., C-LIME, SmoothGrad, Vanilla Gradients) to recover polynomials asymptotically analogous to the sinusoidal case, as every continuous function (in a finite domain) can be written as a sum of sinusoids via a Fourier series representation.
> > > > >
> > > > > **What is not studied when we analyze the explanations method in the LFA framework?**
> > > > >
> > > > > We do not study the following questions as part of our analysis: 1) how to analyze the faithfulness of few other feature attribution methods which are not instances of the LFA framework (e.g, DeconvNet, Grad-CAM, Grad-CAM++, FullGrad, and DeepLIFT) -- See Appendix A.1 for a detailed discussion on this; and 2) how to analyze the faithfulness of other interpretable surrogate models (e.g., shallow decision trees) beyond linear models. While these are interesting directions to explore for future work, our work is the first to theoretically analyze the faithfulness of several popular feature attribution methods (such as LIME, SHAP, Occlusion, C-LIME, Vanilla Gradients, Gradient x Input, SmoothGrad, and Integrated Gradients), making a critical first step towards systematically characterizing the field of post hoc explanations.
> > > > >
> > > > > We hope the aforementioned discussion satisfactorily addresses your questions and concerns. We kindly request you to consider increasing your score.
> > > > >
> > > > > **References**:
> > > > >
> > > > > 1. Zachary Lipton. The mythos of model interpretability. Queue 16, 3 (2018), 31–57.
> > > > >
> > > > > 2. Cynthia Rudin. Stop explaining black box machine learning models for high stakes decisions and use interpretable models instead. Nature Machine Intelligence (2017).
> > > > >
> > > > > 3. Satyapriya Krishna*, Tessa Han*, Alex Gu, Javin Pombra, Shahin Jabbari, Steven Wu, and Himabindu Lakkaraju. The Disagreement Problem in Explainable Machine Learning: A Practitioner’s Perspective. arXiv preprint arXiv:2202.01602 (2022).
> > > > >
> > > > > 4. Marco Ribeiro, Sameer Singh, Carlos Guestin. Why Should I Trust You?": Explaining the Predictions of Any Classifier. ACM SIGKDD Conference on Knowledge Discovery and Data Mining (2016).

---

> > > > > > ### Comment · Reviewer_JzYe · 2022-08-09
> > > > > > **Response from: JzYe**
> > > > > >
> > > > > > I am thankful and appreciate the responses from the authors. Based on the current discussion, I, unfortunately, cannot increase my score at this stage. I believe that to increase my score, the authors need to address the concerns I raised in a more theoretical or empirical manner.

---

> > > > > > > ### Author Response · Authors · 2022-08-09
> > > > > > > **Clarification/response regarding newly requested theory/experiments**
> > > > > > >
> > > > > > > Thank you once again for your responses and active engagement. It helps us immensely. Regarding additional theory or experiments, we are interested in learning what specific theory or experiments you would like to see?
> > > > > > >
> > > > > > > Please note that this request for additional theory or experiments was only raised in comments today / yesterday (i.e., the day of the deadline) and the request does not specify what theory to prove or what experiments to conduct. So, we are unfortunately unable to act on this request at this time. Nonetheless, we will address any specific to-dos in the final version of the paper. We sincerely appreciate all your feedback and engagement.

---

> ### Author Response · Authors · 2022-08-08
> **Any Further Questions?**
>
> Dear Reviewer,
>
> We thank you again for your insightful reviews and thoughtful suggestions. As you are aware, the discussion period is coming to a close on Tuesday, and we would thus request you to consider responding ahead of this deadline to provide us with an opportunity to address any remaining questions.
>
> Thank you again for taking the time to evaluate our work and our responses.
>
> Authors

---

### Official Review · Reviewer_vX68 · 2022-07-06

**Rating:** 8
**Confidence:** 3
**Soundness:** 4 excellent
**Presentation:** 4 excellent
**Contribution:** 3 good

**Summary:**

The authors present a framework for local function approximation explanations (LFA) for generating explanations of black box ML models. They present theoretical results that place several existing explanation methods from the literature within the LFA framework; they provide a variety of theoretical and computational results demonstrating the value of this framework.


**Questions:**

None

**Limitations:**

The authors sufficiently addressed the limitations of their work.

**Strengths And Weaknesses:**

The authors very clearly present their conceptual framework (LFA). Their theoretical results are helpful (and intiutive, though this isn't bad). Their theoretical and computational results clearly demonstrate the utility of LFA.

In the context of explaining ML models, their contributions are a bit narrow: this paper focuses on explanations based on local function approximations, and they use fidelity as a measure of explanation "goodness". But the authors are extremely thorough in exploring this region of ML explanations. Overall, I don't have anything bad to say about this paper.

---

> ### Author Response · Authors · 2022-08-02
> **Response to Reviewer vX68**
>
> We would like to thank the reviewer for the insightful comments. We are glad that the reviewer recognizes the importance, soundness, and excellent presentation of our work and appreciates its contribution to the field. Thank you very much!

---

### Official Review · Reviewer_oxiS · 2022-07-12

**Rating:** 5
**Confidence:** 3
**Soundness:** 3 good
**Presentation:** 3 good
**Contribution:** 3 good

**Summary:**

This work presents a framework called local function approximation that can be used to unify local feature importance explanations (based on gradients and perturbations).

**Questions:**

a) Can you mention a few practical uses of the LFA framework for data scientists or practitioners ?

b) Is it possible to overcome any limitation in any of the explanation methods proposed in the literature through the LFA framework ?

c) If my use case of explainability is debugging a model or identifying bias. Can you elaborate how the LFA framework could help me decide which explanation method I should use for these use cases.

d) Could you please elaborate if the LFA framework could help me evaluate the different explanation methods that it unifies and how?







**Ethics Review Area:**

["I don’t know"]

**Limitations:**

See above.

**Strengths And Weaknesses:**

The overall usefulness of the LFA framework is questionable.

Based on the current work, a framework that unifies perturbation methods and gradient based methods is "good to have" in terms of understanding all the methods together and perhaps designing new methods in future. The idea of characterising explanation methods via model recovery which is also good. However, whether the framework can help practitioners decide or guide which explanation method to use for which dataset is still not clear.

Some of the ideas discussed related to perturbation methods are partially known in AI explainability community. The fact that one could fit a model alternate to ridge regression for LIME or one could do train/test on the perturbed dataset generated by LIME, etc. or may be use an alternate loss function, etc.

---

> ### Author Response · Authors · 2022-08-02
> **Response to Reviewer oxiS - Part 1**
>
> We thank the reviewer for their detailed comments. Below, we address the questions and comments raised by the reviewer.
>
> **Ideas related to perturbation methods partially known -- e.g., one could fit a model alternate to ridge regression for LIME or one could do train/test on the perturbed dataset generated by LIME, etc. or may be use an alternate loss function** : While the reviewer is correct that prior works have explored the idea that one could fit a model alternate to ridge regression for LIME or one could do train/test on the perturbed dataset generated by LIME, etc. or may be use an alternate loss function, our work is the first to establish that eight popular post hoc explanation methods are all performing local linear approximations of the underlying model, and we are the first to mathematically characterize the exact loss functions and local neighborhood definitions corresponding to each of these methods. By doing so, our work not only unifies several state-of-the-art explanation methods via the local function approximation (LFA) framework, but also highlights precisely how these methods differ from each other. We then leverage this characterization to analyze which methods are likely to be effective w.r.t. what kinds of data distributions and underlying models. Such a contribution is critical to the progress of the field of explainable AI as it not only provides a unified lens for comparing and contrasting seemingly diverse methods, but also provides answers to one of the biggest open questions in the field -- which explanation methods should be employed under what conditions?
>
> **Practical uses of the LFA framework for data scientists or practitioners**: Our work can be incredibly helpful to data scientists and practitioners in addressing the following critical questions: 1) which explanation method should be employed to explain a given prediction? and 2) how to develop novel explanation methods which are suited to a given context?. Note that while prior research on explainable AI (e.g., Krishna et. al., 2022; Neeley et. al., 2021) argued that these are critically important questions for practitioners, none of the existing works provide answers to these questions and our work makes the first attempt at addressing these questions that are of critical importance to practitioners. Below, we discuss the specifics of how our work enables us to answer the aforementioned questions.
>
> 1) Prior works (e.g., Krishna et. al., 2022) show that practitioners select among methods in an ad hoc manner, which is concerning because relying on incorrect explanations can lead to harmful consequences. To this end, our work lays out a simple yet effective guiding principle i.e., choose a method which can perfectly approximate the underlying model if/when the model class is the same as the explanation function class. We then analyze the conditions under which various existing methods can perfectly approximate the underlying model (Section 4.2). Based on this analysis, we provide specific recommendations to practitioners for which methods (existing methods or new methods created using the LFA framework) to use in different situations:
>
>    When the input domain is binary-valued, one should use binary noise methods (e.g., LIME, KernelSHAP, Occlusion). When the input domain is continuous, one should use additive continuous noise methods (e.g., SmoothGrad, Vanilla Gradients, C-LIME) or modified multiplicative continuous noise methods (e.g., Integrated Gradients, Gradient x Input). More details about the specific recommendations (Section 4.3, lines 264-274), rationale of these recommendations (Section 4.2, lines 203-249), and empirical validation of the rationale (Section 5.2, lines 315-327, and Figure 2) is provided in the main paper.
>
> 2) When existing explanation methods are not appropriate for a given situation (i.e., when existing methods cannot perform faithful function approximation of a given black-box model in the given input domain), practitioners might want to develop new methods with minimal overhead. Our LFA framework enables practitioners to create new explanation methods very easily. All that they need to do is specify an appropriate interpretable model class, perturbation neighborhood (noise distribution and operation to combine the noise and input), and loss function. For details, please see Section 4.3, lines 251-263.
>
> Therefore, by unifying various explanation methods, the LFA framework 1) guides the choice of which method to use under what conditions, and 2) enables the creation of new methods for a given context when existing methods are not appropriate. The practical guidance to practitioners emerging as a result of the conceptual and theoretical understanding facilitated by the LFA framework is a key contribution of this paper.
>
> (continued)

---

> > ### Author Response · Authors · 2022-08-02
> > **Response to Reviewer oxiS - Part 2**
> >
> >  **Is it possible to overcome any limitation in any of the explanation methods proposed in the literature through the LFA framework?**: The LFA framework is a conceptual and theoretical framework that not only advances our understanding of diverse explanation methods, but also provides ways to modify existing methods to overcome their limitations.
> >
> > For example, LFA framework not only establishes that methods such as Integrated Gradients and Gradient x Input do not recover the weights of the underlying linear model (i.e., do not generate explanations that are perfect model approximations) when the input data is continuous, but also enable us to address this issue readily by just changing the loss function -- See Section 4.2, lines 213-224. This is only made possible because our work establishes that these methods are instances of the LFA framework.
> >
> > In addition, as discussed in our previous answer, LFA framework enables researchers and practitioners to easily develop new explanation methods which can overcome specific limitations of existing methods. All that they need to do is specify an appropriate interpretable model class, perturbation neighborhood (noise distribution and operation to combine the noise and input), and loss function. For details, please see Section 4.3, lines 251-263.
> >
> >
> > **Use case of debugging a model or identifying bias -- how can LFA help?**: Faithfulness of post hoc explanations is a critical aspect that influences any downstream task including model debugging or bias detection. Only if the explanations are correct (i.e., faithful to the underlying model), can one be certain that the downstream tasks which rely on these explanations are drawing the right conclusions. The LFA framework and our work more broadly focuses on characterizing the faithfulness of the explanations generated by various state-of-the-art approaches, and laying out the conditions under which they can perfectly approximate the underlying model. By doing so, it is enabling researchers and practitioners with the knowledge of which explanation methods are likely to be faithful in a given context, and is thereby indirectly improving the quality of various downstream applications.
> >
> > **Does the LFA framework help me evaluate the explanation methods that it unifies?**: Our LFA framework enables the evaluation of faithfulness of various state-of-the-art methods by precisely laying out the conditions under which each method can approximate the underlying model with low error. By doing so, it provides researchers and practitioners with a clear understanding of which methods are likely to be most faithful under different data distributions, definitions of local neighborhoods, and model properties. For instance, in Section 4.2, we leverage our LFA framework to demonstrate the following:
> >
> > When the input data is binary-valued, binary noise methods (e.g., LIME, KernelSHAP, Occlusion) can perfectly approximate the underlying model when it is linear. In contrast, methods which employ continuous noise (e.g., SmoothGrad, Vanilla Gradients, C-LIME, Integrated Gradients, Gradient x Input) do not perfectly approximate the underlying model in case of binary-valued data.
> >
> > When the input data is continuous, additive continuous noise methods (e.g., SmoothGrad, Vanilla Gradients, C-LIME) or modified multiplicative continuous noise methods (e.g., Integrated Gradients, Gradient x Input) can perfectly approximate the underlying model when it is linear. In contrast, binary noise methods (e.g., LIME, KernelSHAP, Occlusion) cannot perfectly approximate the underlying model when the data is continuous.
> >
> > By enabling the aforementioned comparisons and insights, LFA is allowing us to effectively evaluate the faithfulness of various state-of-the-art methods both theoretically and empirically. In contrast, prior work has mostly focused on small-scale empirical analyses of the faithfulness of post hoc explanation methods, and it is unclear if and when such empirical results generalize to other settings.
> >
> > We hope that our response addresses your questions/comments and we hope that you consider increasing your score. Thank you very much!
> >
> > ---
> >
> > **References**:
> >
> > Satyapriya Krishna*, Tessa Han*, Alex Gu, Javin Pombra, Shahin Jabbari, Steven Wu, and Himabindu Lakkaraju. The Disagreement Problem in Explainable Machine Learning: A Practitioner’s Perspective. arXiv preprint arXiv:2202.01602 (2022).
> >
> > Michael Neely, Stefan F. Schouten, Maurits J. R. Bleeker, and Ana Lucic. Order in the Court: Explainable AI Methods Prone to Disagreement. arXiv preprint arXiv:2105.03287 (2021)

---

> ### Author Response · Authors · 2022-08-08
> **Any Further Questions?**
>
> Dear Reviewer,
>
> We thank you again for your insightful review. As you are aware, the discussion period is coming to a close on Tuesday, and we would thus request you to consider responding ahead of this deadline to provide us the opportunity to address any remaining concerns.
>
> Thank you again for taking the time to evaluate our work and our responses.

---

### Official Review · Reviewer_Fevw · 2022-07-12

**Rating:** 5
**Confidence:** 4
**Soundness:** 3 good
**Presentation:** 3 good
**Contribution:** 2 fair

**Summary:**

In this paper, the authors propose a generic framework which encapsulates different local explanation techniques as special cases of their LFA framework (linear function approximation). They introduce the no free lunch theorem here within the perspective of local explanations claiming that no single explanation method can perform local function approximation across all neighborhoods. Some of the key experimental results include comparison between Captum's explanation and the explanation from proposed approach by comparing the proximity of these. In addition, perturbation based tests have also been done to reinforce the notion of the no free lunch theorem

**Questions:**

1) In line with the review above, I would expect authors to add on what insight is this no free lunch theorem providing to the XAI community which was not known before ?

**Ethics Review Area:**

["I don’t know"]

**Limitations:**

Yes

**Strengths And Weaknesses:**

**Strengths**

1) Paper is easy to follow and the authors have supported their claims through benchmarked results. Overall coverage of local explainers within their framework is also reasonably exhaustive.

**Weakness**

1) While the no free lunch theorem here might appear a really novel contribution, I don't find it particularly appealing as papers in the past Dylan et al (Fooling LIME and SHAP - AIES 2020) and several others have demonstrated that neighborhood samplers are always prone to issues such as adversarial attacks, bias, etc. I believe the no free lunch theorem is just a fancier way of presenting something already known in the community.

2) I would have liked to see some more results on how LFA is more robust to issues such as overfitting etc (due to train/dev/test and hyper parameter tuning advantages, etc). Its a simple experiment but can enhance value of the paper possibly.

**Post Rebuttal**

I have changed my score after reading the author response to my queries.

---

> ### Author Response · Authors · 2022-08-02
> **Response to Reviewer Fevw - Part 1**
>
> We thank the reviewer for their insightful comments and feedback. We appreciate the positive comments concerning the exhaustiveness of the paper and the quality of the writing. Below, we address specific questions raised by the reviewer:
>
> **No free lunch theorem -- connections with Slack et. al. (Fooling LIME and SHAP - AIES 2020) and other related works**:  Our no free lunch theorem and more broadly our work are fundamentally quite different from Slack et. al. and other related works. The focus of the no free lunch theorem is not on adversarial attacks on explanation methods or hiding model biases, but rather on quantifying the limits of how well explanation methods can approximate the underlying model w.r.t. various definitions of local neighborhoods.
>
> More specifically, Slack et. al. show that LIME and KernelSHAP rely heavily on input perturbations that are typically out-of-distribution (OOD) samples, and adversaries can exploit this to generate adversarial classifiers which can fool these methods into hiding undesirable biases of the underlying models. On the other hand, our no free lunch theorem establishes that no single state-of-the-art explanation method can perfectly approximate the underlying model across all possible definitions of local neighborhoods. Furthermore, no free lunch theorem is a theoretical result which applies to the eight state-of-the-art explanation methods discussed in this work as well as any other instantiation of the LFA framework, whereas most of the prior works (including Slack et. al.) rely only on empirical results to highlight vulnerabilities of explanation methods.
>
> **No free lunch theorem -- Insight**: The no free lunch theorem (NFL) proves, for the first time, that no state-of-the-art explanation method can perfectly approximate the underlying model across all possible definitions of local neighborhoods. Figure 3 in the main paper demonstrates this, showing that no single method performs well across local neighborhoods characterized by both continuous and binary perturbations.
>
> We believe this result is not well known in the explainable AI community, which contains a multitude of papers proposing novel explanation methods, each claiming to be superior to previous ones unequivocally. A consequence of the NFL theorem is that such claims are provably incorrect for methods that can be written as local function approximations.
>
> Note that, **in addition to the no free lunch theorem, our work makes several other key contributions to address fundamental problems in the literature on local explanation methods**. Our work is the first to establish that eight popular post hoc explanation methods are all performing local linear approximations of the underlying model, and we are the first to mathematically characterize the exact loss functions and local neighborhood definitions corresponding to each of these methods (Table 1, Theorems 1 & 2 in main paper). By doing so, our work not only unifies several state-of-the-art explanation methods via the local function approximation (LFA) framework, but also highlights precisely how these methods differ from each other. We then leverage this characterization to carry out a first-of-its-kind theoretical and empirical analysis to determine which explanation methods are likely to faithfully approximate the underlying models under what conditions (Section 4.2). Such a contribution is critical to the progress of the field of explainable AI as it not only provides a unified lens for comparing and contrasting seemingly diverse methods, but also provides answers to one of the biggest open questions in the field -- which explanation methods should be employed under what conditions?
>
> In **summary**, our work is the first to unify eight popular local post hoc explanation methods by showing that they all perform local linear approximations of the underlying models using different loss functions and definitions of local neighborhoods. We then leverage the aforementioned unification to make one of the first attempts at theoretically and empirically characterizing the faithfulness (i.e., how well each method approximates the underlying model?) of eight state-of-the-art explanation methods under various conditions (e.g., data distributions, model properties etc.).
>
> Please refer to our comment above titled "Response to all reviewers" for further details about our contributions and the significance of our work.
>
> (continued to next comment)

---

> > ### Author Response · Authors · 2022-08-02
> > **Response to Reviewer Fevw - Part 2**
> >
> > **LFA -- robustness to overfitting**: In response to the reviewer's comment, we carried out the suggested experiment. The results of the experiments are provided in Section 1 of “additional_results_rebuttal.pdf” (under Supplementary Materials). Below, we describe the details of this experiment and also discuss the results.
> >
> > Setup: In this experiment, we fit an interpretable model g (linear or logistic regression) to locally approximate an underlying black-box model f (linear/logistic regression, and three neural networks). The experiment compares how well g approximates f when g was trained using a validation set (to tune hyperparameters) versus when g was trained without a validation set (where hyperparameters are selected randomly). We use the WHO and HELOC test sets from the empirical analyses of the main paper (Section 5), and generate explanations for each data point in this test set. To explain a given data point, g is trained on a set of perturbations (either using a validation set or not). Then, g is evaluated on an independent test set of perturbations by comparing the g’s predictions to f’s predictions and calculating the mean absolute error (MAE) of these two sets of predictions. We compare the MAE distribution (which measures g’s error in approximating f) when g is trained with and without a validation set. Results are shown in Figures A and B in Section 1 of  “additional_results_rebuttal.pdf” (under Supplementary Materials)
> >
> > Results: The results of the experiment indicate that g tends to perform better when trained using a validation set to tune hyperparameters, than when trained without a validation set. Thus, the results favor our approach of training g using a validation set to tune hyperparameters. As a matter of principle, the method of training a model by splitting a dataset into train/validation/test splits and using a validation set to tune hyperparameters and reduce overfitting is well-established in machine learning literature, and we simply advocate for its adoption in explanation methods.
> >
> > We hope that our response addresses your questions/comments and we hope that you consider increasing your score. Thank you very much!

---

> ### Author Response · Authors · 2022-08-08
> **Any Further Questions?**
>
> Dear Reviewer,
>
> We thank you again for your insightful reviews and thoughtful suggestions. As you are aware, the discussion period is coming to a close on Tuesday, and we would thus request you to consider responding ahead of this deadline to provide us the opportunity to address any remaining concerns.
>
> Thank you again for taking the time to evaluate our work and our responses.

---

> > ### Comment · Reviewer_Fevw · 2022-08-08
> > **Response Post Rebuttal**
> >
> > I appreciate the efforts the authors have taken to address my concerns and I have raised my score accordingly.

---

### Author Response · Authors · 2022-08-02
**Response to all reviewers**

We thank all the reviewers for their insightful comments. Below, we provide a brief overview of our contributions and the significance of our work. We address additional questions and comments in our responses to individual reviewers.

Prior research has argued that a key fundamental challenge plaguing the literature on post hoc explanation methods (particularly, local feature attribution based explanation methods) is the lack of a coherent and systematic characterization of these methods and their effectiveness [1, 2, 3]. While there exist a plethora of these methods (e.g., LIME, SHAP, Gradient based methods etc.), they all adopt different notions of what constitutes a feature attribution based local explanation, and employ different strategies for constructing these explanations. Therefore, it remains unclear as to how these methods relate to each other, and which methods are likely to be effective under what conditions.

Our work addresses the aforementioned fundamental problem by theoretically and empirically demonstrating that eight popular feature attribution based local explanation methods are essentially performing local linear approximations of the underlying model, albeit, with different loss functions and different notions of local neighborhoods. Our work is the first to mathematically characterize the exact loss functions and local neighborhood definitions employed by these methods (Table 1, Theorems 1 & 2 in main paper). By doing so, our work not only unifies several state-of-the-art explanation methods via the local function approximation (LFA) framework, but also highlights precisely how these methods differ from each other. We then leverage this characterization to carry out a first-of-its-kind theoretical and empirical analysis to determine which explanation methods are likely to faithfully approximate the underlying models under what conditions (Section 4.2). In addition, we also introduce the *no free lunch theorem for explanation methods* to show that no single explanation method can approximate the underlying model with low error across all possible definitions of local neighborhoods (Section 4.1). We believe that the aforementioned contributions are critical to the progress of the field of explainable AI as they not only provide a unifying lens for comparing and contrasting seemingly diverse methods, but also address one of the biggest open questions in the field -- which explanation methods should be employed under what conditions?

In **summary**, our work is the first to unify eight popular local post hoc explanation methods by showing that they all perform local linear approximations of the underlying models using different loss functions and definitions of local neighborhoods. We then leverage the aforementioned unification to make one of the first attempts at theoretically and empirically characterizing the faithfulness (i.e., how well each method approximates the underlying model?) of eight state-of-the-art explanation methods under various conditions (e.g., data distributions, model properties etc.). In contrast, prior work in explainable AI has mostly focused on small-scale empirical analyses of the faithfulness of post hoc explanation methods, and it was unclear if and when the results of such analyses hold and/or generalize broadly.

Lastly, we would like to highlight that we conducted **additional experiments** requested by some of the reviewers as part of our rebuttal and discussions. Results from these experiments can be found in Sections 1 and 2 of “additional_results_rebuttal.pdf” in the Supplementary Materials, and these results reinforce that various state-of-the-art feature attribution based explanation methods perform local linear approximations of the underlying models, and are instances of our LFA framework, and also (re)validate our theoretical results concerning the faithfulness of various explanation methods. More details about these additional experiments and findings can be found in our responses below titled "Response to Reviewer Fevw - Part 2" and "Response to Reviewer JzYe - Part 1".

---

**References**:

[1] Zachary Lipton. The mythos of model interpretability. Queue 16, 3 (2018), 31–57.

[2] Cynthia Rudin. Stop explaining black box machine learning models for high stakes decisions and use interpretable models instead. Nature Machine Intelligence (2017).

[3] Satyapriya Krishna*, Tessa Han*, Alex Gu, Javin Pombra, Shahin Jabbari, Steven Wu, and Himabindu Lakkaraju. The Disagreement Problem in Explainable Machine Learning: A Practitioner’s Perspective. arXiv preprint arXiv:2202.01602 (2022).

---

### Meta-Review · Area_Chair_bHiS · 2022-08-28

**Recommendation:** Accept
**Confidence:** Certain

**Metareview:**

The decision is to accept the paper.

The authors propose a unifying Local Function Approximation framework for characterizing a number of local explanation methods into a single formalism. Using this formalism, the authors provide a no-free-lunch theorem indicating that no explanation method can dominate all others across all perturbation specifications. The authors then provide a guiding principle regarding global recovery of the underlying function when the explanation function class is complex enough, and give several examples where different explanation methods do/do not satisfy this principle. There are also empirical results to confirm the theory, and suggestions for designing new explanation methods based on the theory.

Overall, there was agreement that the LFA framework provides a useful unified perspective on many different explanation methods. Some of the remaining concerns from reviewers could be addressed by discussing the limitations of the framework more explicitly in the main paper. For example, referencing appendix A.1 and the types of methods that are not characterized by this framework; or discussing how the choice of explanation method may also depend on how the particular use case is mapped to neighborhoods etc, which is out of scope here (in the spirit of the current comment about the choice of G). This is mainly a matter of expanding the discussion a bit, which should be feasible with an extra page.

**Award:**

No

---

### Decision · Program_Chairs · 2022-09-14

Accept